# A Novel In Vitro Model of the Bone Marrow Microenvironment in Acute Myeloid Leukemia Identifies CD44 and Focal Adhesion Kinase as Therapeutic Targets to Reverse Cell Adhesion-Mediated Drug Resistance

**DOI:** 10.3390/cancers17010135

**Published:** 2025-01-03

**Authors:** Eleni E. Ladikou, Kim Sharp, Fabio A. Simoes, John R. Jones, Thomas Burley, Lauren Stott, Aimilia Vareli, Emma Kennedy, Sophie Vause, Timothy Chevassut, Amarpreet Devi, Iona Ashworth, David M. Ross, Tanja Nicole Hartmann, Simon A. Mitchell, Chris J. Pepper, Giles Best, Andrea G. S. Pepper

**Affiliations:** 1Department of Clinical and Experimental Medicine, Brighton and Sussex Medical School, Falmer, Brighton BN1 9PX, UK; k2sharp@gmail.com (K.S.); f.a.simoes@bsms.ac.uk (F.A.S.); j.jones2@bsms.ac.uk (J.R.J.); tom_burley@hotmail.co.uk (T.B.); l.stott1@uni.bsms.ac.uk (L.S.); a.vareli@bsms.ac.uk (A.V.); e.m.kennedy@bsms.ac.uk (E.K.); sv274@sussex.ac.uk (S.V.); t.chevassut@bsms.ac.uk (T.C.); i.ashworth1@uni.bsms.ac.uk (I.A.); s.a.mitchell@bsms.ac.uk (S.A.M.); c.pepper@bsms.ac.uk (C.J.P.); a.pepper@bsms.ac.uk (A.G.S.P.); 2Department of Haematology, Brighton and Sussex University Hospital Trust, Brighton BN2 5BE, UK; amarpreet.devi1@nhs.net; 3Department of Haematology, Flinders Medical Centre, College of Medicine and Public Health, Flinders University, Adelaide, SA 5042, Australia; david.ross@sa.gov.au (D.M.R.); giles.best@flinders.edu.au (G.B.); 4Department of Medicine I, Medical Center—University of Freiburg, Faculty of Medicine, University of Freiburg, 79085 Freiburg, Germany; tanja.hartmann@uniklinik-freiburg.de

**Keywords:** acute myeloid leukemia (AML), bone marrow microenvironment, adhesion, CAM-DR, CD44, FAK

## Abstract

Acute myeloid leukemia (AML) is a challenging blood cancer to treat, with only about 24% of patients surviving for 5 years after diagnosis. A key challenge is that AML cells stick to normal cells in the bone marrow (BM), and these BM cells protect them from chemotherapy. The aim of this project is to find drugs that disrupt AML cell adherence to BM cells and release them into the blood, where chemotherapy will be more effective. To achieve this, we have created a model of adhesive BM and shown that it mimics the drug resistance seen clinically. We have used the model as a testing platform for drugs that disrupt AML cell adhesion. We have shown that the combined targeting of CD44 and FAK, using anti-CD44 and the clinical-grade FAK inhibitor defactinib, inhibits the adhesion of the most primitive AML cells that are associated with drug resistance and disease relapse.

## 1. Introduction

AML is characterized by the uncontrolled expansion of myeloid progenitors in the bone marrow (BM) and peripheral blood (PB) and remains a therapeutic challenge due to its aggressive nature and genetic and phenotypic heterogeneity [1,2]. Relapse after initial response to chemotherapy occurs in the majority of patients (~80%) [3] and remains a serious clinical challenge requiring new therapeutic strategies. The estimated median overall survival (OS) of AML is 8.5 months, with a 24% 5-year OS. Nearly 80% of AML patients diagnosed at ≥65 years will die within one year [4]. One of the issues hindering the successful treatment of AML and contributing to disease relapse is leukemic cell retention in the protective niche of the BM [5]. Here, the leukemic cells are surrounded by multiple cell types that promote tumor cell survival, enabling them to evade destruction via systemic therapies, leading to the emergence of drug resistance [6].

The mechanisms that lead to adhesion-mediated protection of leukemic cells in the BM are complex and involve multiple cytokines, chemokines and adhesion molecules. Three cell types that have been identified as important components of the BMME are stromal cells, endothelial cells and osteoblasts [7,8]. Stromal cells have been shown to protect AML cells from spontaneous and drug-induced apoptosis by direct contact [5] and reciprocal NF-κB activation [9]. They induce NF-κB signaling, which leads to the transcriptional activation of anti-apoptotic genes in AML cells, resulting in increased tumor cell survival and resistance to chemotherapy [10]. Stromal cells are also known to enhance the metabolic activity of AML cells, which contributes to the chemo-resistance associated with residence in the BMME [11]. Endothelial cells not only provide an essential nutrient-delivering vascular network for the tumor cells but also play a role in tumor cell adhesion and migration between the BM and circulation [12]. Signaling between leukemic and endothelial cells is bi-directional, as leukemia-mediated endothelial cell activation has been found to upregulate the cell adhesion molecule E-selectin, promoting AML cell quiescence and a chemo-resistant state [13]. The presence of osteoblasts also creates a tumor-promoting niche; AML cells can promote the osteogenic differentiation of stromal cells into osteoblasts [14]. Despite all three cell types having tumor-promoting properties, drugs that target these interactions are not routinely used in AML.

AML and normal hemopoietic stem cells (HSCs) adhere to the BM via a number of mechanisms, including the receptors CXCR4, very late antigen-4 (VLA-4) and the cell surface glycoprotein CD44 [15]. The elevated expression of CXCR4 on AML cells is associated with decreased patient survival [16], and attempts have been made to mobilize leukemic cells out of the protective niche of the BMME using CXCR4 inhibitors such as plerixafor. Clinical trials demonstrated some efficacy of CXCR4 inhibitors; however, not all AML cells are mobilized [17,18]. This indicates that whilst the CXCR4 interaction with its ligand CXCL12 is important, other molecules play a role in the adhesion of AML cells in the protective niche of the BMME. VLA-4 is an integrin dimer composed of CD49d (α4) and CD29 (β1) and has also been associated with chemo-resistance in AML [19]. VLA-4 binds to the extracellular matrix (ECM) protein fibronectin and VCAM-1 expressed by stromal and endothelial cells. In a murine model of AML, survival was significantly prolonged in mice that were given VLA-4-blocking antibodies in combination with cytarabine than those receiving cytarabine alone [19]. CD44 is a glycosylated class-I transmembrane protein that binds to glycosaminoglycan, hyaluronic acid (HA, an important component of the BM ECM), osteopontin (OPN), collagens and matrix metalloproteinase [20]. HA and CD44 expression have been implicated in AML, with Hartmann et al. demonstrating the importance of CD44-induced activation of VLA-4 in AML adhesion [21]. The expression of CD44 was essential for AML engraftment in a mouse model [22], and targeting CD44 reduced AML stem cell re-population in a serial transplant model [23]. Interestingly, CD44 has been shown to be a key player in drug resistance of other hematological malignancies such as multiple myeloma [24,25]. To target BM resident chemo-resistant AML cells, a better understanding of the biological interactions between leukemic cells and the hemopoietic niche is needed. In view of the above, CXCR4, VLA-4, CD44 and E-selectin represent promising targets for further investigation.

The aims of this study were two-fold: firstly, to develop a multicellular in vitro co-culture system that mimics the adhesive AML-BMME; and secondly, to identify the most effective agent(s) for blocking the adhesion of AML cells and to establish whether they can reverse cell adhesion-mediated cytarabine resistance.

## 2. Methods

Full details of all methods are provided in the Appendix A and a table of the reagents used in Appendix A.

### 2.1. Cell Culture

**Cell lines**: KG1a, OCI-AML3, HS-5, hFOB 1.19 and HUVECs were purchased from ATCC (Manassas, VA, USA) and DSMZ (Braunschweig, Germany) and grown in the recommended media (OCI-AML3 and KG1a: RPMI 1640 + 20% FBS, 1% Penicillin/Streptomycin and 1% L-glutamine (all Merck, St. Louis, MO, USA); HS-5: DMEM (Fisher, Pittsburgh, PA, USA) + 10% FCS, 1% Penicillin/Streptomycin and 1% L-glutamine; HUVECs: Medium 199 (Fisher) + 20% FCS, 1% Penicillin/Streptomycin and 1% L-glutamine; hFOB 1.19: DMEM/F-12 MIX (Fisher) without phenol red + 10% FBS, 2.5% L-glutamine and 0.3 mg/mL G418 (Fisher)). hTERT-transfected HUVECs were previously generated and characterized over multiple passages in the Pepper lab [26] and were gown on plate pre-incubated for 30 min with 0.2% Gelatin (Merck).

**Primary cells**: Patients with a new AML diagnosis were recruited (Appendix A). PB samples and/or BM samples were obtained with full ethical approval and following informed consent in accordance with the Declaration of Helsinki.

**Co-culture and cell adhesion quantification**: Co-culture assays were performed on 12-, 24- and 96-well plates with a consistent cell density and stroma/AML cell ratio. Full details of this, the drugs used, isolation/identification and the counting of non-adhered and adhered AML cells are in the Appendix A.

### 2.2. Immunophenotyping

Labeling with fluorescent antibodies was carried out according to the manufacturer’s instructions using the clones listed in Appendix A. Briefly, cells were initially blocked using Biolegend (Biolegend, San Diego, CA, USA) cell staining buffer for 30 min. They were then collected into FACS tubes at densities of 3 × 10^5^–1 × 10^6^ in 100 µL staining buffer or PBS. The appropriate surface antibodies were added, and the tubes were vortexed. Cells were then incubated for at least 20 min in the dark at 4 °C. Post-staining, the cells were washed twice with 1 mL of PBS or staining buffer and centrifuged gently at 300× *g* for 5 min. After two washes, the stained cells were resuspended in 300 µL of PBS for immediate analysis using a CytoFLEX LX flow cytometer (Beckman Coulter, Brea, CA, USA). AML cells were identified and gated as CD45^high^CD73^low^ and stromal cells as CD45^low^CD73^high^. The cell expression of other antigens was measured using median fluorescent intensity (MFI).

Staining for pFAK was performed using the True-Phos™ kit (Biolegend, San Diego, CA, USA) and PE-anti-pFAK (BD Biosciences, Franklin Lakes, NJ, USA) as per the manufacturer’s instructions. The MFI values for p-FAK were determined in gated CD45+ CD73-AML cells using the CytoFLEX LX flow cytometer.

### 2.3. AML Cell Isolation

The EasySep™ human CD45 Positive Selection Kit and an EasySep™ magnet (both STEMCELL Technologies UK Ltd., Cambridge, UK) were used to isolate the AML cells as per the manufacturer’s instructions. To ensure all AML cells were treated in an identical fashion, both adhered and non-adhered AML cells were selected.

### 2.4. Measurement of In Vitro Apoptosis

Selected adhered, non-adhered and monoculture AML cells were stained using BioLegend Annexin V FITC and binding buffer following the manufacturer’s instructions. Briefly, up to 1 × 10^6^ cells were resuspended in 100 µL of Annexin V binding buffer, and subsequently, 2.5 µL of Annexin V FITC (Biolegend) was added to the cell suspension, and the cells were incubated in the dark at RT for 15 min. Each sample was acquired on the CytoFLEX LX, and analysis was performed using CytExpert 2.4 software. The experiment was performed in triplicate.

### 2.5. RNA Sequencing

Full details are provided in the Appendix A.

### 2.6. Statistical Analysis and Synergy

All statistical analyses were performed using GraphPad Prism 9.5.1 (GraphPad Software, San Diego, CA, USA). In all cases, the normal distribution of the data was assessed using the Shapiro–Wilk test. Assuming that this assumption was met, univariate comparisons were made using a *t*-test for paired observations by comparing the means of the replicates between the two groups. If the means between >2 groups were compared, one-way ANOVA with Dunnett’s correction (which compares every mean to a control mean) for multiple comparisons was used. Toxicity data from the drug treatments were used to produce sigmoidal dose–response curves. From these interpolated LC_50_ values (the concentration of drug required to kill 50% of the cells) were calculated following normalization and constraint (bottom value = 0 and top value between 0 and 100). The expected drug combination responses were calculated based on a ZIP reference model using SynergyFinder (SynergyFinder software (version 3.0, https://synergyfinder.fimm.fi (accessed on 15 February 2023)). Deviations between the observed and expected responses with positive and negative values denote synergy and antagonism, respectively. For the estimation of outlier measurements, the cNMF algorithm implemented in SynergyFinder was utilized. For all statistical tests, a confidence level of 95% was used, and therefore, *p* values < 0.05 were deemed significant.

## 3. Results

### 3.1. The “BM Adhesion System” (BMAS)

Although many in vitro co-culture systems have been established for AML, none have measured AML cell adhesion in a multicellular, bone marrow-relevant system. Stromal cells, endothelial cells and osteoblasts are abundant and, importantly, functionally significant in the BMME [7,8], potentially all using different cocktails of adhesion receptor/ligands for adhesion. Therefore, to create a model that mimics the heterogeneous adhesive and protective BMME, and which can be used as a drug-testing platform, we used HS-5 (stromal cells), HUVECs (endothelial cells) and hFOB 1.19 (osteoblasts). We found all three to be adhesive to AML cells to variable degrees (Appendix A), with different expression levels of common adhesion markers (Appendix A, e.g., HUVECs are 100% CXCR4 positive, whereas HS5s do not express CXCR4), highlighting the importance of using a multicellular system. Subsequently, we used HS5, hFOB 1.19 and HUVECs (stroma mix) plated in equal proportions (1:1:1, the BMAS) (Appendix A).

To negate the need for trypsinization and AML cell selection of adhered cells prior to counting, the adhesion of AML cells was measured indirectly by counting the number of AML cells still in suspension in the presence of the BMAS (non-adhered). For each experiment, an equal number of AML cells was plated in monoculture and in co-culture with the BMAS. The BMAS non-adhered AML cells are presented as a percentage of AML cells in monoculture. AML cell lines were chosen to represent distinct AML phenotypes. KG1a cells are BM-derived AML cells representing a more primitive AML disease (containing a CD34^hi^CD38^dim^ stem-like sub-population) [27]. In contrast, OCI-AML3 cells are PB-derived and represent a more differentiated disease [28] (Appendix A). Despite their divergent characteristics, both AML cell lines exhibited strong adhesion on the BMAS in comparison to those cultured alone (KG1a non-adhered: 23%, *p* = 0.0019; OCI-AML3 non-adhered: 28%, *p* = 0.004) (Figure 1a,b).

To confirm that adherence in the BMAS mimics the protective nature of the BMME, we measured the viability of BMAS adhered (following detachment using TryPLE) and non-adhered AML cells. The adhered cells were both significantly less apoptotic (OCI-AML3 = 25% apoptotic versus 42%, *p* = 0.001 and KG1a = 6.8% apoptotic versus 14%, *p* = 0.001 in adhered versus non-adhered, respectively; Figure 1c). These results confirm that the in vitro BMAS recapitulates some of the functionally important aspects of the AML BMME, namely its adhesive nature and promotion of tumor cell viability.

### 3.2. The BMAS Model’s Cell Adhesion-Mediated Drug Resistance (CAM-DR) and Soluble Factor-Mediated Drug Resistance (SFM-DR)

To establish whether the BMAS also models AML chemo-resistance, we performed cytarabine dose–response assays and measured apoptosis with Annexin V after 72 h. This was performed on non-adhered, adhered and monoculture AML cells plus the stromal cells. In the BMAS co-cultures, the non-adhered AML cells were removed initially, and then the remaining adhered AML cells and stromal cells were removed using TryPLE. Cells were then stained using antibodies to CD45 and CD73 to identify the AML and stromal cells prior to the addition of Annexin V. Strikingly, in both OCI-AML3 and KG1a cells, the LC_50_ (cytarabine dose that kills 50% of cells) was not achieved in the adhered AML cells even at the highest dose of 10 µM, indicating a strong chemotherapy resistance effect created by adhesion in the BMAS: adhered LC_50_ > 10 µM. For the BMAS non-adhered cells, the LC_50_ values were OCI-AML3 7.38 µM and KG1a 4.57 µM and for monoculture cells OCI-AML3 1.87 µM and KG1a 2.81 µM (Figure 2). As the non-adhered cell LC_50_ was higher than that of monoculture cells, soluble factors in the BMAS must also contribute to cytarabine resistance, but not as much as cell adhesion (Figure 2). Importantly, cytarabine had no significant effect on the stroma mix cells (Figure 2).

Our in vitro system, therefore, successfully emulates both the cell adhesion mediated drug resistance (CAM-DR) and soluble factor-mediated drug resistance (SFM-DR), as seen in the BMME. These results also suggest that prevention of AML adhesion within the BMME could diminish chemotherapy resistance.

### 3.3. Anti-CD44 Most Effectively Reduces AML Adhesion

We next sought to see if we could prevent AML cell adhesion in the BMAS. Our phenotyping studies identified that both AML cell lines expressed high levels of the adhesion molecules CXCR4, CD49d and CD44 (Appendix A). Therefore, we tested whether adhesion could be prevented using several commercially available blocking agents to CXCR4 (plerixafor [17,18] and ONO-7161), CD49d (natalizumab) and CD44 (anti-CD44). We also tested an E-selectin-blocking antibody as it is expressed on the 21% of HUVEC cells in our co-culture (Appendix A) and has previously been suggested as important in AML cell adhesion [13,29,30,31].

Testing the impact of each blocking agent across a range of concentrations revealed that treatment with anti-CD44 significantly reduced adhesion at all doses in both cell lines (Figure 3a,b). Compared to untreated controls, 5 µg/mL anti-CD44 treatment resulted in a 1.8-fold reduction in the adhesion of OCI-AML3 cells (Figure 3a) and a 3.6-fold reduction in adhesion in KG1a cells (Figure 3b). Importantly, this concentration of anti-CD44 was more effective than the highest concentration of CXCR4-blocking agents (plerixafor and ONO-7161), CD49d (natalizumab) and E-selectin (anti-E-selectin) (Figure 3c). Individual dose–response plots for each of these agents can be seen in Appendix A.

We next tested the effectiveness of anti-CD44 on primary BM- and PB-derived AML cells. Similarly to the cell lines, AML cells from all patient samples tested adhered to the BMAS (mean non-adhered: 24.8%, ±16.98% [range 6–64%]). Compared to untreated cells, anti-CD44 reduced the adhesion of both BM- and PB-derived AML cells up to 4-fold with a mean of 1.3-fold (*p* < 0.001, *n* = 10) and 1.5-fold (*p* < 0.0001, *n* = 15), respectively (Figure 3d). In summary, our co-culture system revealed that blocking CD44 was effective at blocking AML cell adhesion in both cell lines and primary cells. We hypothesized that if CD44 is an important adhesion marker for BM retention, BM expression of CD44 will negatively correlate with the PB white blood cell (WBC) count, as AML cell retention in the BM will reduce AML cells in the PB. Using eight BM-derived primary samples, the surface expression of adhesion molecules on blasts was assessed for correlation with the PB WBC count (taken at the time of BM biopsy). The PB WBC count was negatively correlated with CD44 expression (*r* = −0.7, *p* = 0.044; Figure 3e). In contrast, no correlation was observed with CXCR4 (*r* = −0.23, *p* = 0.584) or CD49d (*r* = 0.007, *p* = 0.987), supporting the hypothesis that CD44 is one of the most important molecules in BM retention of AML cells. To confirm this, we compared previously published BM CD44 mRNA levels with the percentage of AML blasts in the BM [32] and found a significant positive correlation (*r* = 0.32, *p* < 0.0001; Figure 3f).

### 3.4. Anti-CD44 Treatment Potentiates Anti-Tumor Effects by Reducing CAM-DR

We next combined anti-CD44 with chemotherapy in our in vitro culture system to investigate whether modulating adhesion through CD44 can reduce CAM-DR. Compared to no-antibody controls, the addition of 5 µg/mL anti-CD44 resulted in significantly more apoptotic AML cells using three different cytarabine concentrations (Figure 4a): a 1.39–1.66-fold increase in apoptotic OCI-AML3 cells and a 1.44–1.8-fold increase in apoptotic KG1a cells. A representative dot plot is shown in Figure 4b, where the combination treatment resulted in more non-adhered AML cells and, therefore, more Annexin V^+^ apoptotic AML cells. Figure 4c shows the residual co-culture after all the non-adhered cells were removed and reveals that the combination resulted in strikingly few viable AML cells still adhered to the stroma mix, which itself remained viable and adhered to the plastic plate.

These findings were confirmed using primary AML cells, where combination treatment resulted in up to a 1.9-fold increase in apoptotic cells compared to cytarabine alone in both PB- and BM-derived AML cells (Figure 4d,e). Interestingly, we found that for 13/15 individual patient samples, more apoptosis was seen in the wells containing both drugs (5 µg/mL anti-CD44 with 5 µM cytarabine) than when we added together the total amount of apoptosis seen in the wells with cytarabine alone with that in anti-CD44 alone: mean ± SD apoptotic cell number of 3036 ± 2482 [combination] versus 1928 ± 1772 [sum], *p* = 0.009 (Figure 4f and Appendix A). Importantly, in the presence of anti-CD44 and cytarabine, the number of apoptotic primary AML cells correlated with their pre-experiment expression of CD44 (*r* = 0.73, *p* = 0.002; Appendix A).

### 3.5. Transcriptomic Analysis Identified FAK as a Determinant of Persistent Adhesion Following Treatment with Anti-CD44

Despite anti-CD44 treatment being the most effective at preventing AML adhesion, some AML cells (using both cell lines and primary AML cells) remained persistently adhered and viable, even in the presence of the highest dose of anti-CD44 and high (micromolar) concentrations of cytarabine. This reflects the heterogeneity of AML, even within a cell line. To identify novel adhesion targets, we isolated and performed RNA sequencing (RNA-seq) on persistently adhered cells and non-adhered cells after anti-CD44 treatment and analyzed it using the online software BioJupies (https://maayanlab.cloud/biojupies (accessed on 15 February 2023)), as described in the Appendix A. Unsupervised clustering of RNA-seq data showed a transcriptional signature associated with persistent adhesion, which was largely consistent across cell lines (Figure 5a). Differential expression analysis was performed separately for each cell line and identified 215 common differentially upregulated genes in the adhered cells compared to non-adhered (Figure 5b). Most differentially expressed genes were upregulated (486 for OCI-AML3 and 364 for KG1a), with only a few genes showing downregulation (Figure 5c). KEGG pathway enrichment analysis identified the over-representation of the focal adhesion kinase (FAK) signaling pathway in the adhered cells of both cell lines, suggesting that targeting this pathway may be therapeutically beneficial for overcoming persistent adhesion in the presence of anti-CD44 (Figure 5d). The RNA-seq results were validated with qRT-PCR (Appendix A) and uploaded to the EMBL-EBI data repository (accession number E-MTAB-14052).

While the expression of FAK itself remained unchanged, measuring FAK phosphorylation (pFAK) in the BMAS in the presence and absence of anti-CD44 showed an increase in FAK signaling in adhered versus non-adhered AML cells. As before, the number of adhered AML cells was much less in the presence of anti-CD44, but pFAK was consistently higher in the adhered AML cells (MFI 3133 ± 94) compared to non-adhered (MFI 2052 ± 34) and monoculture (MFI 1661 ± 99) supporting the use of this signaling pathway for adhesion (Figure 5e).

### 3.6. Defactinib Potentiates Anti-CD44 Treatment in Preventing AML Cell Adhesion to the BMME

As the transcriptomic analysis highlighted the FAK pathway as a potential mechanism of AML adhesion, we attempted to overcome persistent adherence by targeting this pathway in combination with anti-CD44.

Defactinib, a potent dual and reversible ATP-competitive inhibitor of FAK and PYK2 [33], is currently in phase II clinical trials for the treatment of several solid tumors [34,35]. We initially tested the potential of defactinib to reduce pFAK in AML cells and found that after an 18 h incubation followed by 3 h on the BMAS (with and without anti-CD44), defactinib reduced pFAK in both adhered and non-adhered AML cells (*p* = 0.039; Figure 5f). We next tested defactinib as an anti-adhesion therapy in our model and compared to untreated KG1a cells found that it significantly reduced their adhesion (mean reduction of 1.5-fold; *p* < 0.04) (Appendix A). Surprisingly, defactinib did not have the same effect on the OCI-AML3 cell line (Appendix A). Although elements of the FAK pathway were over-expressed in adhered cells from both cell lines, the expression of FAK (PTK2) itself was five times higher in the more primitive KG1a cells compared to OCI-AML3 cells, which may explain their increased sensitivity to defactinib. To investigate whether a higher expression of FAK was associated with a more primitive phenotype in AML, we compared previously published CD34 and FAK (PTK2) mRNA levels in AML cell lines and two AML patient cohorts [32,36,37]. Despite the challenges of patient-to-patient heterogeneity across multiple RNA-seq datasets, a subtle but consistent correlation between FAK and CD34 expression was seen (*p* ≤ 0.01 for cell line data and q-value < 0.0001, following Benjamini–Hochberg FDR correction, for both patient cohorts). These data suggest that blocking FAK may be most effective in CD34^high^-expressing cases of AML (Appendix A). In support of this, we found that defactinib-sensitive KG1a cells had significantly higher protein expression of CD34 compared to OCI-AML3 cells (Appendix A). It is worthy to note that the adhered AML cells on the BMAS system also showed significantly higher CD34 expression than non-adhered cells (Appendix A).

When defactinib was combined with 5 µg/mL anti-CD44 treatment, the combination of the two prevented the adhesion of KG1a cells up to a mean of 4.4-fold compared to untreated controls. This was substantially better than either drug alone, with the drugs combining in an additive fashion (mean Bliss score = 1.63) (Figure 6a,b). This indicates that dual targeting of these two pathways may represent a promising therapeutic strategy. To test this further, we co-cultured primary AML cells with autologous stromal cells grown out from the BM aspirates of three patients. The patient samples selected were known to have >15% CD34^high^ AML cells in their BM aspirate. Similar to the cell line BMAS model, a significantly higher proportion of non-adhered CD34^+^ cells were present in the samples treated with all concentrations of either anti-CD44 (up to 3.8-fold) or defactinib (up to 3.1-fold) alone. However, in the samples treated with combinations of the two drugs, the number of non-adhered CD34+ cells were significantly higher than in the control or single-drug-treated samples for all three patients and at all dose combinations tested. The most effective concentration combination was 2.5 µg/mL anti-CD44 and 2.5 µM defactinib for all three patients, with changes of 14.2-, 8.98- and 12.19-fold (compared to untreated samples), respectively. Representative scatter plots from one patient sample are shown in Figure 6b, showing substantially more non-adhered CD34+ and CD34- cells in the presence of both drugs, and combined data for the three patients are in Figure 6c. Furthermore, the dose combinations were highly additive in all three patient samples at all concentrations (Bliss scores > 1). At 2.5 µg/mL of anti-CD44 + 2.5µM of defactinib, the Bliss scores indicated synergy (Bliss score > 10) in all patients (18.74, 11.59 and 12.93) (Figure 6d). Taken together, these results suggest that defactinib and anti-CD44 treatment combine synergistically to reduce adhesion in patients with CD34^+^ AML.

## 4. Discussion

One of the issues hindering the successful treatment of AML patients and contributing to disease relapse is leukemic cell adhesion and retention in the protective niche of the BMME. Here, the leukemic cells are surrounded by other cell types, including stromal cells, endothelial cells and osteoblasts, that promote their survival by enabling them to evade destruction by both the immune system [38,39,40,41,42] and intra-vascular therapies [10,11], ultimately leading to the emergence of drug resistance. The disruption of these AML cell interactions in the BMME has the potential to make them far more susceptible to standard therapies.

We have developed a novel, robust and physiologically representative BM adhesion system, the BMAS. We used three cell types that are known to be abundant in the BM [7,8]—stromal cells (HS-5), endothelial cells and osteoblasts—to create a highly adhesive system, which we tested using both AML cell lines and primary cells. The system modeled the adhesiveness of the BM seen in vivo [21], with a large percentage of AML cells adhered, being more viable than their non-adhered counterparts. Furthermore, the BMAS was also shown to successfully recapitulate both CAM-DR and SFM-DR, again consistent with that seen in the AML BM [43,44]. Importantly, the system has utility as a drug-testing platform.

To identify potentially druggable adhesion targets, we measured the levels of adhesion molecules known to have commercially available blocking agents—namely CD44, CXCR4, CD49d and E-selectin—and found all except E-selectin to be highly expressed on AML cells. In keeping with the data from clinical trials [17,18], targeting CXCR4 with plerixafor reduced the adhesion of a small proportion of AML cells in the BMAS, and similar results were achieved using ONO-7161. Similarly, blocking CD49d with a clinical-grade inhibitor commonly used for the treatment of multiple sclerosis [45] mobilized just a small number of AML cells. Even with the highest dose of anti-E-selectin, no effect was observed.

The potential importance of CD44 in AML adhesion and initiation has already been highlighted by Hartmann et al. [21]. Furthermore, anti-CD44 is available as a clinical-grade humanized monoclonal antibody (RG7356), has been trialed as a therapy for CD44-expressing solid tumors [46] and been found to be safe and well tolerated as a monotherapy in a phase 1 trial for AML [47]. However, its utilization as a BM anti-adhesion agent, to reverse CAM-DR and sensitize AML cells to chemotherapy, has not been previously explored. Here, we show that blocking CD44 in our BMAS is effective in preventing AML adhesion and that combining this with cytarabine significantly increases AML cell apoptosis. Importantly, the combination of 5µM cytarabine and 5 µg/mL anti-CD44 increased apoptosis of primary AML cells more than the additive effect of either alone, supporting the hypothesis that forcing AML cells out of the protective BM niche will render them more susceptible to standard therapies. This finding is supported in a study by Bjorklund et al., who showed that blocking CD44 in vitro reduced adhesion and sensitized myeloma cells to lenalidomide [25]. Furthermore, an in vivo study utilized an anti-CD44 antibody to target liposomes containing cytarabine to AML cells. Although the study concluded that the reduced WBCs in both BM and PB were due to liposomal targeting, it is possible that the blockade of CD44 and, thus, disruption of AML adhesion were also occurring [48].

Even though anti-CD44 treatment was the most effective blocking agent in our study, some AML cells still retained the ability to adhere. Transcriptional profiling of anti-CD44-treated persistently adhered AML cells identified them to be enriched in the FAK pathway, and the measurement of FAK phosphorylation confirmed an increase FAK signaling, indicating that the inhibition of this in combination with anti-CD44 could be a promising therapeutic strategy. Defactinib (dual FAK and PYK2 inhibitor) is a clinically utilized drug that has been shown to have potential in models of chronic lymphocytic leukemia in vitro [49]. In December 2022 (estimated completion 2025), a new phase I clinical trial started looking at the combination of defactinib and decitabine/cedazuridine (ASTX727) in AML, Myelodysplastic Syndrome and Chronic Myelomonocytic Leukemia (CMML) patients (CELESTIAL-MDS trial). We showed that defactinib reduced adhesion in the more primitive KG1a CD34^high^ cell line and had an additive effect when combined with anti-CD44 treatment, preventing adhesion better than either agent alone. Interestingly, defactinib showed no effect in the more differentiated CD34^low^-expressing OCI-AML3 cells. An analysis of previously published CD34 and FAK (PTK2) mRNA levels in AML cell lines and two AML patient cohorts showed a subtle but consistent correlation between FAK and CD34 [32,36,37], suggesting that defactinib might be most effective in more primitive CD34^high^ AML cells. Importantly, it is widely accepted that it is stem-like populations of cells that are associated with CAM-DR and relapse [50], and we show here that BMAS-adhered AML cells have a more primitive CD34^high^ phenotype. The observation that defactinib preferentially targets these cells is an interesting finding of this study.

Importantly, the adhesion blockade seen in KG1a cells was replicated in CD34^high^ primary AML cells on an autologous BMME. The striking reduction in adhesion and synergy between the two drugs in all three of the primary samples tested highlights the combination of anti-CD44 and defactinib as a promising and novel therapeutic mobilization strategy. The next step would be to test the ability of this combination to chemo-sensitize AML cells in pre-clinical in vivo models. This will initially be determined in CD34^high^/CD44^high^ AML, where the data presented here suggest that it will be the most effective. Since anti-CD44 is available as a recombinant immunoglobulin G1 humanized monoclonal antibody [46] and drug repositioning strategies always result in faster development to trial rates, the addition of this agent to the defactinib and ASTX727 strategy has exciting implications for this largely incurable disease.

## 5. Conclusions

Adhesion of leukaemic cells in the bone marrow microenvironment (BMME), play an important role in the resistance of AML to current therapeutic agents. Here we created a multi-cellular, physiologically relevant, in vitro model of the adhesive and chemo-protective AML BMME (BMAS) and demonstrated that it can recapitulate the CAM-DR seen clinically. Using this model we show that: 1. CD44 is a far more potent AML adhesion target than CXCR4, CD49d and E-selectin. 2. The addition of anti-CD44 reduces AML cell adhesion and substantially increases cytarabine induced apoptosis. 3. AML cells that remain persistently adhered, despite treatment with anti-CD44, show increased expression of the focal adhesion kinase pathway (FAK) and FAK phosphorylation. 4. Dual targeting of CD44 and FAK (using anti-CD44 and the clinical grade FAK inhibitor defactinib) synergistically inhibit adhesion of the most primitive CD34^high^ AML cells that are associated with CAM-DR and relapse.

## Figures and Tables

**Figure 1 cancers-17-00135-f001:**
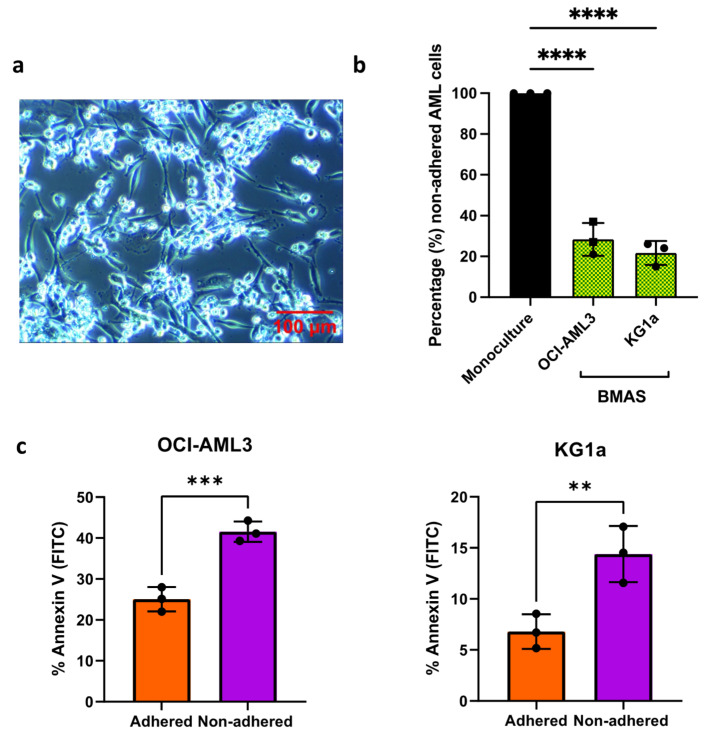
The optimized BMAS. (**a**) The 10× magnification light microscopy images of co-culture. KG1a cells (bright round) attached to BMAS cells (darker and elongated). Images were captured using an Olympus CKX41 microscope, with a micropix camera and Tsview 7 version 7.1 software. (**b**) The number of non-adhered AML cells (mean ± SD) when co-cultured with ratio of 1:1:1 hFOB1.19/HS-5/HUVEC, KG1a cells (*n* = 3) and OCI-AML3 cells (*n* = 3) as a percentage of the number in monoculture. Significance determined using a one-way ANOVA, following the Shapiro–Wilk test for normality. (**c**) Percentage Annexin V-positive (apoptotic) adhered versus non-adhered OCI-AML3 (*n* = 3) and KG1a (*n* = 3) cells. Significance determined using paired *t*-test, following the Shapiro-Wilk test for normality. Significance determined using one-sample *t*-test, following the Shapiro-Wilk test for normality. **** *p* ≤ 0.0001, *** *p* ≤ 0.001 and ** *p* ≤ 0.01.

**Figure 2 cancers-17-00135-f002:**
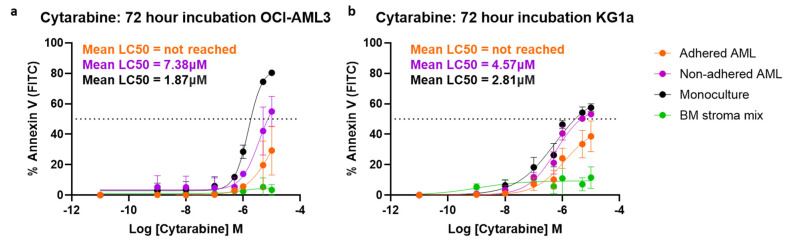
BMAS-modeled CAM-DR and SFM-DR following 72-h of co-culture. Cytarabine dose response curves (72 h). (**a**) OCI-AML3 cells (*n* = 3) and (**b**) KG1a cells (*n* = 3) were treated with increasing doses of cytarabine for 72 h and viability measured using Annexin V staining and flow cytometry. Sigmoidal dose response curves were plotted (mean ± SD).

**Figure 3 cancers-17-00135-f003:**
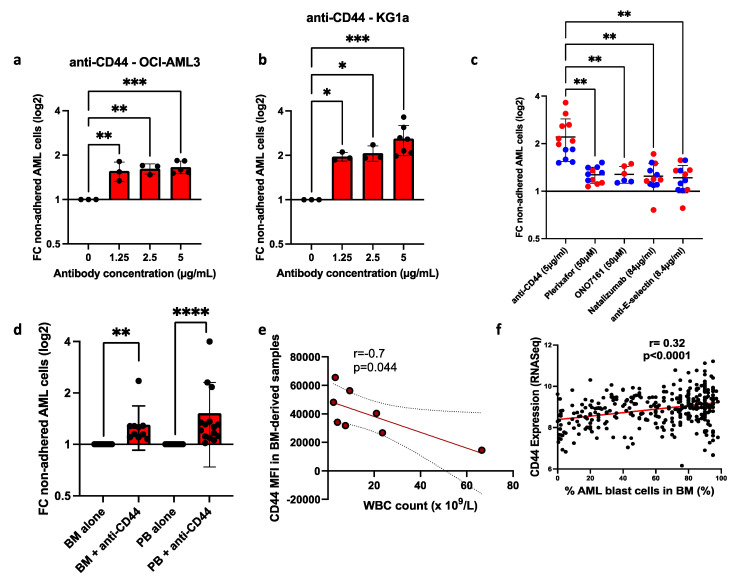
Anti-CD44 treatment was the most effective in blocking AML cell adhesion in cell lines and primary cells. Incubation of AML cells with anti-CD44 antibody. (**a**) Compared to the untreated sample, the fold change (FC) in non-adhered AML cells (mean ± SD) when treated with increasing doses of anti-CD44 and co-cultured for 3 h on the BMAS in (**a**) OCI-AML3 and (**b**) KG1a. Significance determined using one-way ANOVA and Dunnett’s multiple-comparisons test for comparing every mean to a no-treatment control equal to 1. (**c**) A comparison between the best dose for each drug tested, showing anti-CD44 was the most effective (OCI-AML3 blue dots and KG1a red dots). Significance determined using Welch’s ANOVA with Dunnett’s multiple comparisons. (**d**) Compared to the untreated sample, the fold change (FC) in non-adhered primary AML cells (mean ± SD) when treated with 5 µg/mL of anti-CD44 in BM (*n* = 10) and PB (*n* = 15) samples. Primary AML cells were identified using a full AML panel and patient-specific phenotyping data provided by the diagnostic laboratory. A representative panel and gating strategy for primary AML cells can be found in Appendix A. Significance determined using one-sample Wilcoxon, following the Shapiro–Wilk test for normality. Results are compared to a no-treatment control equal to 1. (**e**) Correlation of CD44 expression (median fluorescent intensity [MFI]) in BM-derived samples (*n* = 8) with PB WBC count on samples taken at the same time. Correlation was determined using Pearson’s correlation, with the 95% confidence interval shown as dotted lines. (**f**) Correlation of BM AML cell CD44 mRNA with the percentage of AML blast cells in the BM from BEATAML2 [32]. Correlation was determined using Pearson’s correlation with 95% confidence intervals. **** *p* ≤ 0.0001, *** *p* ≤ 0.001, ** *p* ≤ 0.01 and * *p* ≤ 0.05.

**Figure 4 cancers-17-00135-f004:**
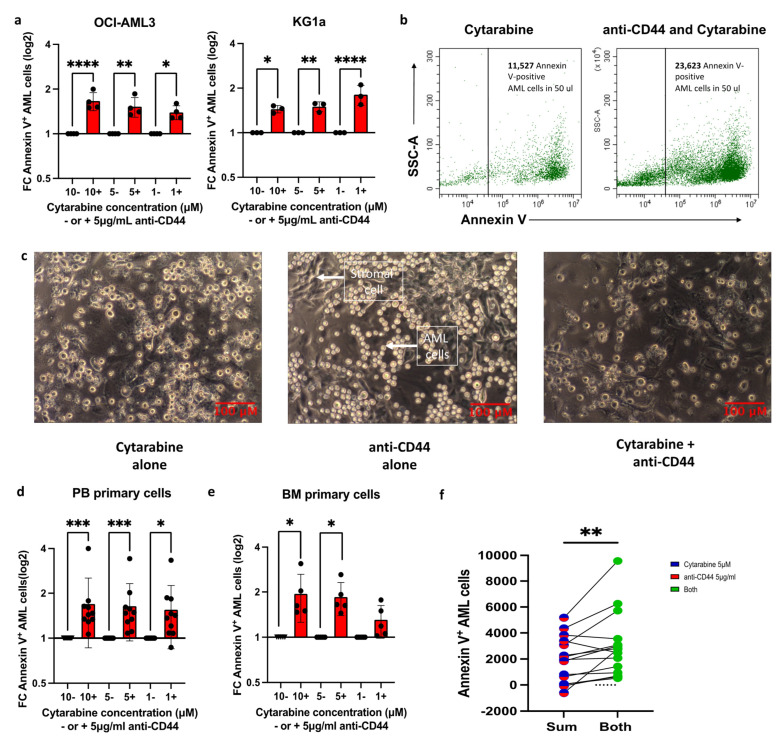
The combination of anti-CD44 with cytarabine can overcome CAM-DR. AML cells were incubated with three different concentrations of cytarabine ± pre-treatment with 5 µg/mL anti-CD44 (**a**) For each cytarabine concentration (10 µM, 5 µM and 1 µM), the fold change (FC) in the number of Annexin V-positive AML cells (mean ± SD) in the presence of anti-CD44 is compared to its absence (the absence of anti-CD44 is normalized to 1): OCI-AML3 (*n* = 3) and KG1a cells (*n* = 3). Significance determined using one-way ANOVA and Dunnett’s multiple-comparisons test comparing every mean to a no-treatment control equal to 1 following the Shapiro–Wilk test for normality (**b**) Representative dot plot of Annexin V staining of non-adhered KG1a cells following treatment with 1 µM cytarabine alone (left) or 1 µM cytarabine with 5 µg/mL anti-CD44 (right). Although the proportion of apoptotic cells is similar in both, there are far more non-adhered KG1a cells in the presence of anti-CD44 (right) and, therefore, far more apoptotic KG1a cells. (**c**) Representative 10× magnification light microscopy images (scale bar represents 100 µm) of the co-culture following treatment with 5 µM cytarabine alone (left), 5 µg/mL anti-CD44 alone (middle) or both (right). Elongated darker cells are the BMAS, which adhered to the base of the well and were unaffected by chemotherapy and anti-CD44 treatment. Some persistently adhered AML cells (round with dark center) were observed following cytarabine treatment alone (left image). More bright, shiny AML cells are seen in the presence of anti-CD44 alone (middle image), but when cytarabine is added to anti-CD44, a marked reduction in the number of adhered AML cells was observed (right image). Images were captured using an Olympus CKX41 microscope, a micropix camera and Tsview 7 version 7.1 software. (**d**,**e**) For each cytarabine concentration (10 µM, 5 µM and 1 µM), the fold change (FC) in the number of Annexin V-positive AML cells (mean ± SD) in the presence of anti-CD44 is compared to its absence (the absence of anti-CD44 is normalized to 1) using (**d**) PB-derived samples (*n* = 10) and (**e**) BM-derived samples (*n* = 5). Significance was determined using the Kruskal–Wallis test and Dunnett’s multiple-comparisons test (**d**) and one-way ANOVA test and Dunnett’s multiple-comparisons test (**e**), following the Shapiro–Wilk test for normality. (**f**) Number of Annexin V-positive AML cells (in 50 µL) following treatment with 5 µM of cytarabine alone added to that in 5 µg/mL anti-CD44 alone in primary AML cells. The sum of their individual effects (red/blue column on left) is compared to their combined effect when cells were treated with both agents simultaneously (green column on right). Results for each individual patient are shown in Appendix A). Each sample represents a biological repeat (*n* = 15). Significance was determined using a paired *t*-test, following the Shapiro–Wilk test for normality. **** *p* ≤ 0.0001, *** *p* ≤ 0.001, ** *p* ≤ 0.01 and * *p* ≤ 0.05.

**Figure 5 cancers-17-00135-f005:**
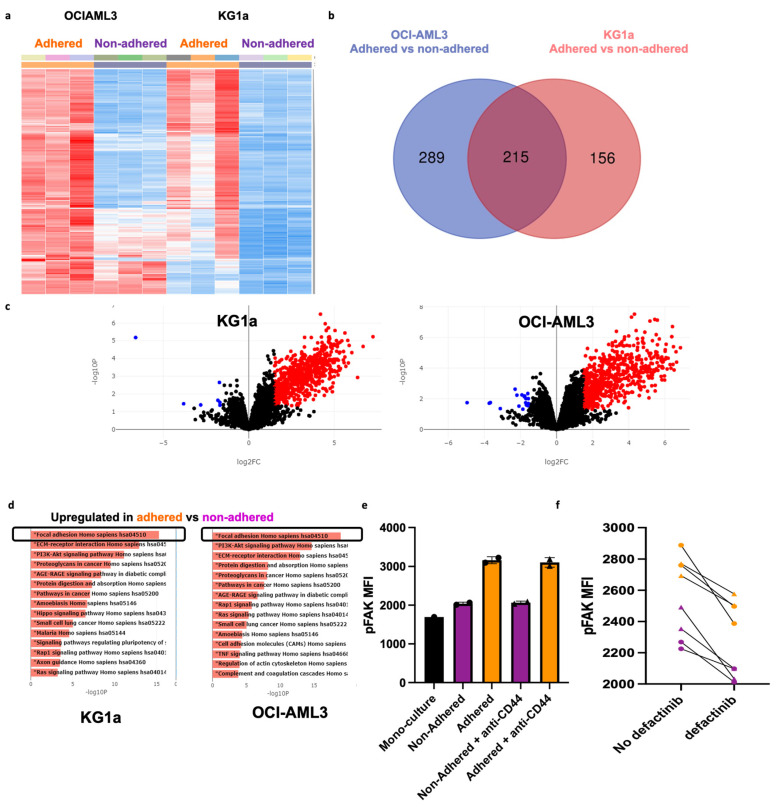
Transcriptomic analysis identified the FAK signaling pathways as the top determinant of persistent adhesion following treatment with anti-CD44. (**a**) Unsupervised hierarchical clustered heatmap for each sample in the RNA sequencing dataset. Every row of the heatmap represents a single gene, every column represents a sample, and every cell displays normalized gene expression values. (**b**) Venn diagram summarizing the overlap between differentially expressed genes between adhered versus non-adhered AML cells after each cell line was analyzed separately. The left circle (blue) represents the genes differentially expressed in OCI-AML3-adhered cells compared non-adhered cells. The right circle (red) represents the genes differentially expressed in KG1a-adhered cells compared to non-adhered cells. (**c**) Volcano Plot displaying the log2-fold changes of each gene, calculated by performing a differential gene expression analysis. Every point in the plot represents a gene. Red points indicate significantly upregulated genes, and blue points indicate downregulated genes for OCI-AML3 (right; 486 upregulated and 18 downregulated) and KG1a (left; 364 upregulated and 7 downregulated) cells. The thresholds used for this analysis were log2FC ≥1.5 and an adjusted *p* ≤ 0.05. (**d**) Pathway enrichment analysis (KEGG pathways) for OCI-AML3 (right) and KG1a (left) cells. The x-axis indicates the −log10 (*p*-value) for each term. Significant terms are highlighted in bold. (**e**) MFI of pFAK in monoculture, adhered and non-adhered KG1a cells in the presence (triangle points) and absence (circle points) of 5 µg/mL anti-CD44. (**f**) pFAK MFI in adhered (orange triangles and circles) and non-adhered (purple triangles and circles) KG1a cells in the presence and absence of 5 µM of defactinib.

**Figure 6 cancers-17-00135-f006:**
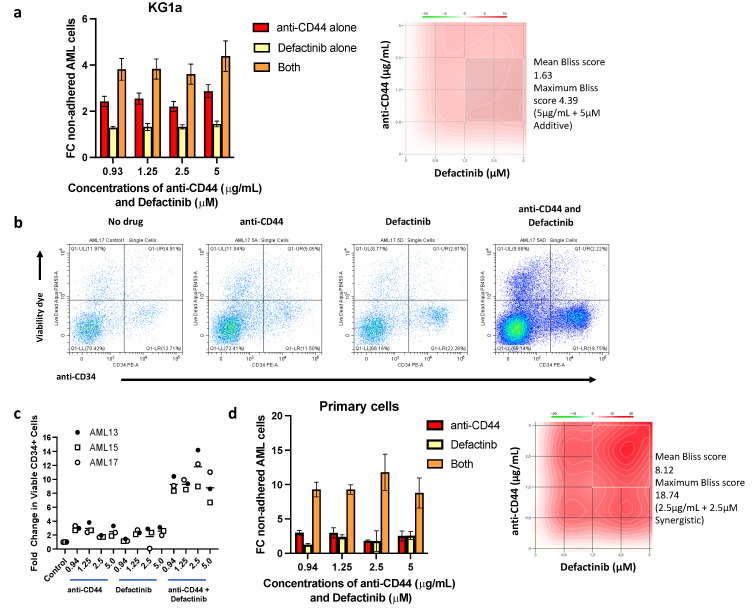
Defactinib in combination with anti-CD44 is additive/synergistic in preventing AML cell adhesion. (**a**) FC in non-adhered KG1a AML cells (mean ± SD) when treated with increasing doses of anti-CD44 alone, increasing doses of defactinib alone or in combination. Synergy plot (right) was generated using SynergyFinder software (version 3.0, https://synergyfinder.fimm.fi (accessed on 15 February 2023)), showing an additive mean Bliss score of 1.627 (>1 = additive) and maximum of 4.39 (5 µg/mL + 5 µM). Results are compared to the no-treatment control, which is equal to 1. (**b**) Representative scatter plots of no drug, anti-CD44, defactinib and the combination of both, showing the proportions of total and CD34^+^ non-adhered primary AML cells after 2 min of acquisition on a Cytoflex S flow cytometer. This shows substantially more viable non-adhered CD34^+^ and CD34^−^ AML cells in the presence of both anti-CD44 and defactinib than no drug or either alone. (**c**) Individual FC (compared to no drug) in viable non-adhered primary AML cell numbers (mean ± SD) when treated with increasing doses of anti-CD44, defactinib or both and co-cultured for 3 h with a confluent layer of autologous stromal cells. Different concentrations of anti-CD44 alone and defactinib alone versus the combination of both were determined using a one-way ANOVA, and the results are tabulated in Appendix A; all comparisons were significant. (**d**) Combined FC in non-adhered primary AML cells (*n* = 3, mean ± SD) when treated with increasing doses of anti-CD44 alone, increasing doses of defactinib alone or in combination. A representative synergy contour plot for patient AML13 (right) was generated using SynergyFinder and shows a mean Bliss score of 8.12 (>1 = additive) and a synergistic maximum of 18.74 (2.5 µg/mL + 2.5 µM; >10 = synergistic).

## Data Availability

The datasets generated during and/or analyzed during the current study are available in the EMBL-EBI data repository, accession number E-MTAB-14052.

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
