# Peer review of "A Novel In Vitro Model of the Bone Marrow Microenvironment in Acute Myeloid Leukemia Identifies CD44 and Focal Adhesion Kinase as Therapeutic Targets to Reverse Cell Adhesion-Mediated Drug Resistance"

_cancers, 2025, doi:10.3390/cancers17010135_

Round 1
Reviewer 1 Report
Comments and Suggestions for Authors
The manuscript titled “A novel in-vitro model of the bone marrow microenvironment in AML identifies CD44 and Focal Adhesion Kinase as therapeutic targets to reverse cell adhesion-mediated drug resistance” identified anti-CD44 and defactinib as a therapeutic combination to release AML cells from the chemoprotective AML BMME. The manuscript is interesting.
However, there are following points which need to be addressed:
1. The authors need to provide scales bares for all the photographic/IHC figures.
2. Please provide details/guidelines for patient ethical consent followed in the study.
3. Please provide details of Conflicts of Interest.
3. Please provide the detailed culture conditions for cell lines in method section.
4. Please provide catalogue numbers/sources of all the kits/chemicals/reagents used in manuscript.
5. Please provide details of statistical analysis employed in the study in method section.
6. Please provide details of Annexin V staining protocol and flow cytometry details used in the study. This information is missing from the current manuscript.
7. Please provide details of Percentage non-adherent AML cells in Fig.1b
8. Define MFI for Fig. 3e.
9. Please provide details of Transcriptomic analysis in fig5.
10. The manuscript requires significant attention specifically in method and material section to improve punctuation, grammar and readability.
Reviewer 2 Report
Comments and Suggestions for Authors
The original article “A novel in-vitro model of the bone marrow microenvironment in AML identifies CD44 and Focal Adhesion Kinase as therapeutic targets to reverse cell adhesion-mediated drug resistance” reported that CD44 and FAK inhibition were potential therapeutic target, and release CAM-DR in AML cells. This manuscript is very interesting, but FAK has already reported as therapeutic targets in AML. Meanwhile, it is a new insight that CD44 and FAK inhibition released CAM-DR in AML cells treated with cytarabine. Therefore, the authors should focus the effect of CD44 and FAK inhibition for CAR-DM more.
For instance, another anti-leukemia therapeutics combined with CD44 and FAK inhibition can be investigated similarly with cytarabine.
Additionally, whether CD44 and FAK inhibition change cell cycle of AML cells or not might be one of another question. These results might suggest that CD44 and FAK inhibition affect not only cell cycle dependent therapeutic, such as cytarabine, but also cell cycle independent therapeutic.
Reviewer 3 Report
Comments and Suggestions for Authors
This is a nice, concise paper demonstrating that CD44 neutralization and FAK inhibition might enhance chemotherapy effects over AML cells. Authors underline that both strategies are already clinically available, and their findings might be quickly translated into a proof-of-principle validation clinical trial.
Authors should discuss the main limitations of their current findings, ie the lack of in vivo data in preclinical models supporting the evidence collected here only in vitro. Also, they should better discuss how to select AML patients that are more likely to benefit from the anti-CD44/defactinib chemo-sensitization. Would they suggest CD44 and/or FAK expression levels?
Reviewer 4 Report
Comments and Suggestions for Authors
This is a very well written manuscript with interesting data. The authors have co-cultured osteoblasts, stromal cells and endothelial cells to generate a bone marrow adhesion model system. Using this system, they showed that sensitivity to cytarabine was reduced. This reduced sensitivity could be reversed by combining with agents that inhibit AML cell interactions with bone marrow cells. Finally, they identified that FAK phosphorylation in CD34+ cells played a role in AML cell adherence even in the presence of anti-CD44 antibodies. Addressing the following points may help clarify further:
Please include details about what the authors mean by “non-adherent cells” in the Results section.
For Fig. 2, please clarify if trypsinization was performed for analyzing adherent cells and BMAS cells. How were the 2 cell types distinguished?
Annexin V positive population was propidium iodide negative?
What was the percentage of non-adherent cells in primary cells? Was it comparable to the cell lines used? As data is presented as fold change, this information is lost.
Fig. 3F, are the data for percentage of blasts in peripheral blood for these patients available? It would be interesting to observe if there is a negative correlation between BM blast percentage and peripheral blood blast percentage.
BLISS score greater than 10 is usually considered synergistic. Please elaborate if a different methodology was used to calculate the score.
Please discuss why there were no differences in the effect of BMAS on BM or PB derived blasts.
Discuss why 1:1:1 ratio was chosen for the BMAS. And if any other ratios were tried? Does this ratio signify the physiological situation?
Round 2
Reviewer 1 Report
Comments and Suggestions for Authors
In the updated manuscript “A novel in-vitro model of the bone marrow microenvironment in AML identifies CD44 and Focal Adhesion Kinase as therapeutic targets to reverse cell adhesion-mediated drug resistance,” the authors have successfully addressed all previous concerns. The manuscript now convincingly presents innovative perspectives on anti-CD44 and defactinib as a promising therapeutic combination to release AML cells from the chemoprotective AML BMME. Therefore, I recommend this article for publication.
Reviewer 2 Report
Comments and Suggestions for Authors
This manuscript was revised well following the reviewer comments, and so suitable for acceptance.